*Review Article*

# Mechanism of autophagy initiation by transmembrane selective autophagy receptors

Elias Adriaenssens [ID] [1,2] ✉ & Sascha Martens [ID] [1,2] ✉

## Abstract

**Selective autophagy ensures the targeted degradation of damaged or surplus cellular components, including organelles, thereby safeguarding cellular homeostasis. This process relies on selective autophagy receptors (SARs) that link specific cargo to the autophagy machinery. These receptors exist in two distinct forms: soluble SARs that are recruited to the cargo on demand, and transmembrane SARs that are stably embedded in the membranes of organelles they target. While both receptor types converge on the same autophagy core machinery, they differ in how they recognize cargo, are regulated, and recruit this machinery to the site of degradation. In this review, we explore the unique challenges and strategies associated with transmembrane SARs, including how their activity is suppressed under basal conditions and activated in response to stress. We compare their mode of action with that of soluble SARs, highlight key differences in kinase regulation, including the roles of TBK1, ULK1, CK2, and Src, and discuss emerging models of autophagy initiation. We further highlight fundamental principles of organelle-selective autophagy and identify open questions that will guide future research.**

**Keywords** Quality Control; Selective Autophagy; Mitophagy; ER-phagy; Autophagosome
**Subject Category** Autophagy & Cell Death

## Introduction

The regulated degradation of cellular material is fundamental for maintaining homeostasis and enabling cells to adapt to changing physiological conditions. While the proteasome fulfils an essential role in degrading individual proteins, the selective clearance of larger protein aggregates or organelles—such as mitochondria, the endoplasmic reticulum (ER), peroxisomes, and lysosomes—plays an equally crucial role (Jayaraj et al, 2020; Pohl and Dikic, 2019). This process is mediated by selective types of autophagy, a set of pathways that distinguishes dysfunctional or superfluous components from functional ones and ensures their targeted delivery to lysosomes for degradation.

Macroautophagy (hereafter autophagy ("self-eating")) is a conserved cellular process that enables the degradation and recycling of cytoplasmic material. During autophagy, a flattened membrane cisterna, called a phagophore, forms and expands around the incipient cargo, and eventually closes to form an autophagosome. This vesicle then fuses first with late endosomes to form an amphisome and subsequently with a lysosome or directly with lysosomes, where its contents are broken down by hydrolytic enzymes (Fig. 1A). This multi-step pathway is a highly orchestrated process that requires the activity of the core autophagy-related (ATG) machinery: a set of proteins characterized by their involvement in the biogenesis of autophagosomes (Chang et al, 2021; Melia et al, 2020; Mizushima et al, 2011; Nakatogawa, 2020). This machinery is comprised of the upstream-acting ULK complex, composed of the ULK1/2 kinase, ATG13, ATG101 and FIP200 (Ganley et al, 2009; Hara and Mizushima, 2009; Hosokawa et al, 2009; Jung et al, 2009; Mercer et al, 2009; Yan et al, 1998), the class III phosphatidylinositol 3-phosphate kinase complex 1 (PI3KC3-C1), consisting of the VPS34 lipid kinase, VPS15, Beclin 1, ATG14L (Itakura et al, 2008; Matsunaga et al, 2009; Sun et al, 2008; Zhong et al, 2009), and NRBF2, a regulatory fifth subunit that enhances lipid kinase activity and promotes complex dimerization (Young et al, 2016). In addition, the machinery includes vesicular compartments containing the ATG9A transmembrane protein (Young et al, 2006), the bridge-like ATG2 lipid transfer proteins (Velikkakath et al, 2012), the typically downstream acting WD repeat protein Interacting with Phosphoinositides (WIPI) proteins (Proikas-Cezanne et al, 2004) and the ATG8 lipidation machinery including the ATG16L1 complex. The ATG8 proteins are a family of ubiquitin-like proteins comprising six members in human cells (LC3A, LC3B, LC3C, GABARAP, GABARAPL1, GABARABL2), which are conjugated to the headgroups of the membrane lipids phosphatidylethanolamine (PE) and phosphatidylserine (PS) by the E1-like ATG7, the E2-like ATG3 and the E3-like ATG12–ATG5-ATG16L complex. The latter complex is itself formed through the action of a second ubiquitin-like conjugation system (Kabeya et al, 2000; Mizushima et al, 2003; Mizushima et al, 1998).

According to a prominent model, autophagosome biogenesis is initiated by establishing membrane contact sites that form between the ER and ATG9A-containing compartments (Chang et al, 2021; Melia et al, 2020) (Fig. 1B). Close association of the ER with phagophores has been observed also for transmembrane SARs (Yamashita et al, 2025). Subsequently, lipids flow from the ER via ATG2 proteins into these ATG9A compartments, resulting in

[1]Max Perutz Labs, Vienna Biocenter Campus (VBC), Dr. Bohr-Gasse 9/Vienna Biocenter 5, 1030 Vienna, Austria. [2]University of Vienna, Max Perutz Labs, Department of Biochemistry and Cell Biology, Dr. Bohr-Gasse 9/Vienna Biocenter 5, 1030 Vienna, Austria. ✉E-mail: elias.adriaenssens@univie.ac.at; sascha.martens@univie.ac.at

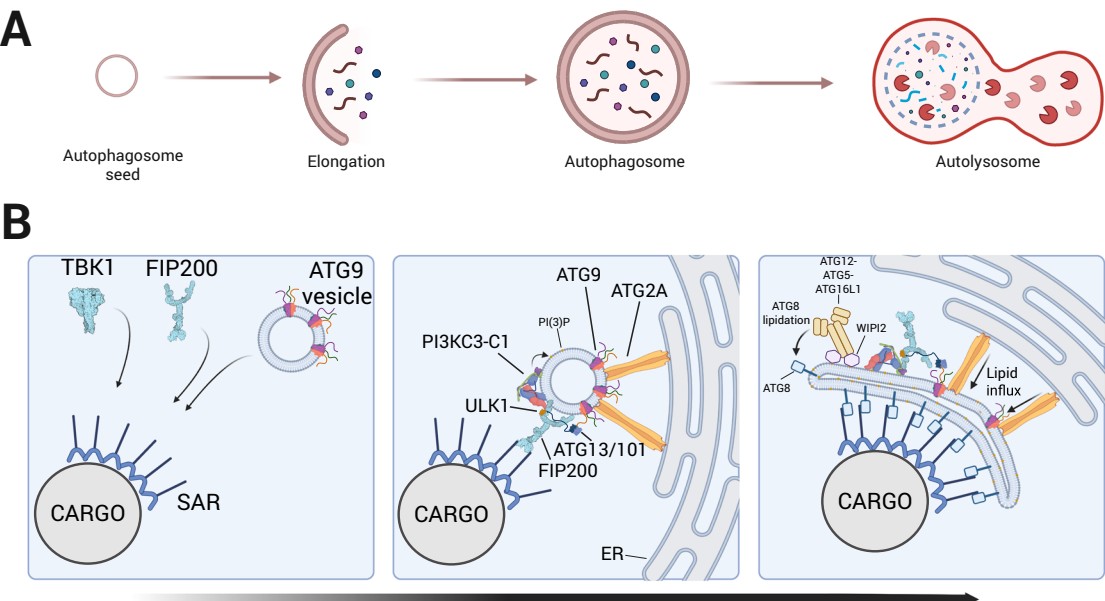

**Figure 1. Schematic overview of the molecular machinery driving autophagosome biogenesis.**

(A) Autophagy enables the degradation and recycling of cytoplasmic material. During this process, ATG9-positive compartments serve as a pre-autophagosomal seed that expands into a flattened membrane cisterna, termed the phagophore, which engulfs cargo and closes to form a double-membrane autophagosome. The autophagosome subsequently fuses with a lysosome, where its contents are degraded by hydrolytic enzymes. (B) In selective autophagy, soluble SARs recognize specific cargo and most SARs recruit the ULK complex directly, composed of ULK1/2, ATG13, ATG101, and FIP200, while OPTN directly recruits TBK1 as well as ATG9A. FIP200 acts as a central scaffold connecting cargo to the early autophagy machinery and ATG9A-positive vesicles through a web of interactions, including the direct interaction between the HORMA domains of ATG13/101 and the ATG9A C-terminal IDR. At membrane contact sites with the endoplasmic reticulum (ER), the lipid transfer protein ATG2A facilitates lipid influx from the ER into the expanding phagophore. Local phosphatidylinositol 3-phosphate (PI3P) production by the class III phosphatidylinositol 3-kinase complex I (PI3KC3-C1; containing VPS34, VPS15, Beclin 1, and ATG14L) promotes recruitment of WIPI proteins, such as WIPI2. This in turn recruits the ATG12–ATG5–ATG16L1 complex, which catalyzes the lipidation of ATG8 family proteins—including the LC3 (LC3A, LC3B, LC3C) and GABARAP (GABARAP, GABARAPL1, GABARAPL2) subfamilies—onto the phagophore membrane. These orchestrated molecular events drive phagophore expansion and closure, ultimately yielding a mature autophagosome.

phagophore expansion. Local phosphatidylinositol 3-phosphate (PI3P) production by the PI3KC3-C1, which is stimulated by the ULK complex, recruits the WIPI proteins including WIPI2. WIPI2 in turn recruits the ATG16L1 complex catalyzing ATG8 lipidation on the growing phagophore. The lipidated ATG8 subsequently recruit further factors of the autophagy machinery and allow selective autophagy receptors (SARs) to attach specific cargo to the membrane. Finally, the ESCRT (Endosomal Sorting Complexes Required for Transport) machinery—a conserved set of membrane-remodeling complexes best known for their roles in multivesicular body formation by cutting and reshaping membranes—appears to convert the phagophore into the sealed double-membrane autophagosome (Chang et al, 2021; Melia et al, 2020).

While autophagy was first described as a non-selective response to starvation, it has become clear that cells do indeed use it to remove specific structures, such as damaged mitochondria, invading bacteria, or protein aggregates (Gatica et al, 2018; Lamark and Johansen, 2021). SARs are a metaphorical seam that runs through this selectivity. They mark targets for degradation, recruit the autophagy machinery to them and act as molecular bridges between the cargo and the nascent autophagosome (Adriaenssens et al, 2022) (Fig. 1B). SARs themselves fall into two major categories: soluble SARs and transmembrane SARs. Whereas most

soluble SARs are recruited to ubiquitylated cargo only when needed, transmembrane SARs are constitutively present on organelle surfaces, anchored through one or more transmembrane helices. Although both types engage similar components of the autophagy initiation machinery, they differ in how their activity is regulated and, in some cases, how they recruit the autophagy machinery.

## Diverging strategies for detecting and marking cargo material

Soluble SARs such as SQSTM1/p62, NDP52, OPTN, TAX1BP1, and NBR1 are not permanently associated with a particular organelle or structure. Instead, they can be quickly recruited to cargo on demand. This recruitment is typically mediated by the ubiquitylation of cargo material (although, here, exceptions exist) (Adriaenssens et al, 2022; Goodall et al, 2022; Lamark and Johansen, 2021). Upon their recruitment, SARs then attract components of the autophagy machinery, initiating autophagosome biogenesis (Lamark and Johansen, 2021; Yamano and Youle, 2020). During this process, the exquisite specificity of the ubiquitylation machinery allows a relatively small number of SARs to monitor and

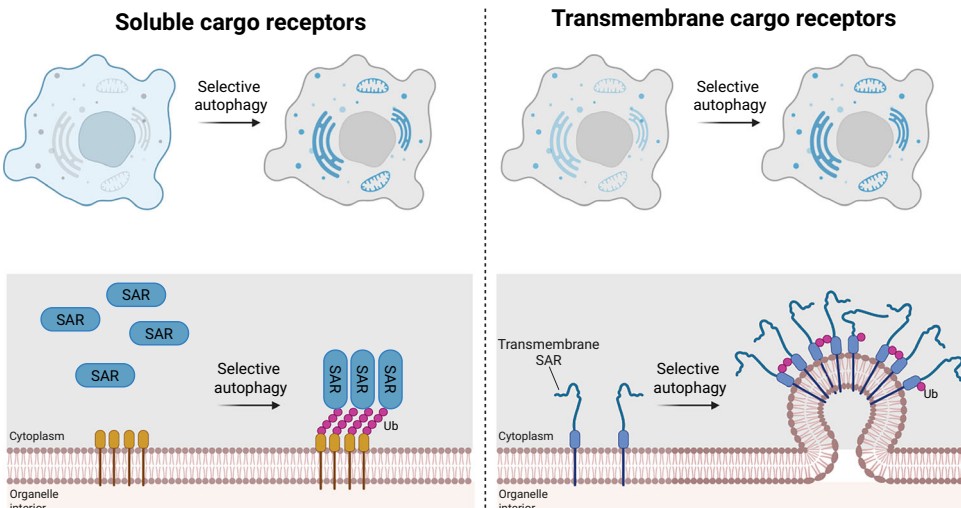

**Figure 2. Activation of soluble versus transmembrane SARs.**

Left, soluble SARs such as SQSTM1/p62, NDP52, OPTN, TAX1BP1, and NBR1 are cytosolic proteins that are recruited to cargo on demand. Their recruitment is commonly driven by ubiquitin (Ub) marks on damaged or misfolded cargo, enabling rapid and flexible targeting of diverse substrates. Upon recruitment, these SARs concentrate autophagy initiation factors at the cargo site to initiate autophagosome biogenesis. Right, transmembrane SARs such as NIX, BNIP3, FAM134B, and CCPG1 are stably integrated within organelle membranes. To prevent unwanted autophagy activation, these SARs require additional layers of control. Selective autophagy can be triggered by transcriptional upregulation (e.g., hypoxia-driven induction of NIX/BNIP3), post-translational modifications such as ubiquitination, or SAR clustering. ER-phagy SARs further promote local membrane bending and scission to excise a portion of the organelle for autophagic removal. While some SARs are transcriptionally induced, others—such as FAM134B—are constitutively present and rely primarily on clustering and post-translational regulation, providing SAR-specific modes of control over autophagosome formation.

target a wide variety of cellular substrates. The relatively high abundance of some of these SARs, reaching concentrations of 400 nM for p62 when averaged over the cell volume (Cho et al, 2022), further enhances this system's responsiveness, enabling rapid adaptation to proteotoxic or organellar stress.

While most soluble SARs recognize ubiquitin-tagged cargo, some operate independently of ubiquitin by directly engaging specific cargoes. In fact, the first SAR identified, the yeast Atg19 protein, directly binds to its prApe1 cargo (Scott et al, 2001). In mammalian cells, the soluble SAR NCOA4 binds ferritin heavy chain subunits and mediates their autophagic degradation (ferritinophagy), a process that releases stored iron ions (Mancias et al, 2014). Similarly, the recently characterized SAR IRGQ targets misfolded MHC class I molecules for lysosomal degradation by interacting with ATG8 proteins, thereby preventing their aberrant surface expression and modulating immune recognition (Herhaus et al, 2024). Another example is glycophagy, a process that regulates glucose homeostasis as glycogen is bound by STBD1 and recruited for autophagic degradation (Jiang et al, 2010; Koutsifeli et al, 2022). These examples illustrate the use of selective autophagy in specialized quality control functions that are independent of classical damage-associated tagging of the cargo with ubiquitin. Such pathways can enable the cell to fine-tune nutrient availability through highly selective and conditionally activated autophagic responses. The list of soluble ubiquitin-independent SARs is probably not yet complete; for example, conceptually, similar strategies might exist for other metabolites or metal-binding complexes, such as zincosomes—cytoplasmic zinc-enriched vesicular structures (visualized with zinc-sensitive dyes) that serve in zinc storage and mobilization (Wellenreuther et al, 2009) (although

dedicated SARs for these potential substrates have yet to be defined).

The spatial flexibility and transient, fast recruitment of soluble SARs contrasts with the fixed localization of transmembrane SARs, which are restricted by their embedding in the cytosol-facing surface of organelles such as mitochondria and the ER (Fig. 2). This membrane integration imposes unique regulatory challenges: how can the cell prevent these relatively immobile SARs—such as NIX and BNIP3 on mitochondria (Daido et al, 2004; Novak et al, 2010), or FAM134B (Forrester et al, 2019; Khaminets et al, 2015; Mochida et al, 2015), FAM134A/C (Kumar et al, 2021; Reggio et al, 2021), and CCPG1 (Smith et al, 2018) on the ER—from triggering unwanted degradation of otherwise healthy organelles?

SAR clustering caused by organelle fragmentation is likely a crucial trigger (Foronda et al, 2023; González et al, 2023). In the case of mitochondria and peroxisomes, both in yeast and mammals, the organelle fission machinery physically interacts with the autophagy machinery and is required for efficient mitophagy or pexophagy, as damaged organelles are broken up into smaller units suitable for selective removal (Chen et al, 2016; Fukuda et al, 2023; Mao et al, 2013; Wu et al, 2016). Mitochondrial fragmentation may enable the spatial restriction of mitophagy SARs to damaged subdomains, thereby preventing global activation across the network. A similar principle underlies the selective accumulation of PINK1 on the surface of depolarized mitochondria, as opposed to its constitutive degradation when mitochondria are healthy (Jin et al, 2010; Narendra et al, 2010; Yamano and Youle, 2013). Interestingly, the mitophagy SARs NIX and BNIP3 appear also to be continuously synthesized and targeted for proteasomal degradation (but this time by the SCF–FBXL4 ubiquitin ligase complex, which resides on the mitochondrial surface) (Cao et al,

2023; Elcocks et al, 2023; Nguyen-Dien et al, 2023). During the degradation of NIX and BNIP3, the phosphatase PPTC7 functions as an adapter bridging the SARs and the SCF–FBXL4 ubiquitin ligase complex (Nguyen-Dien et al, 2024; Niemi et al, 2023; Sun et al, 2024; Wei et al, 2024; Xu et al, 2025). Under homeostatic conditions, therefore, these SARs do not accumulate on healthy mitochondria and mitophagy is silenced. Upon specific stimuli—such as hypoxia or iron chelation—this turnover is suppressed, and SARs accumulate (Allen et al, 2013). In addition to this post-translational regulation, NIX and BNIP3 transcripts become more abundant under hypoxic conditions, primarily through the activation of the hypoxia-inducible factor 1 (HIF-1) pathway, further amplifying their accumulation on mitochondria during stress (Allen et al, 2013; Bruick, 2000; Kothari et al, 2003; Sowter et al, 2001). How NIX or BNIP3 accumulate on discrete mitochondrial subdomains remains unclear, but elegant imaging experiments have shown that BNIP3 forms clusters, and these clusters recruit the autophagy machinery to initiate mitophagy (Wei et al, 2024). Dimerization may facilitate the clustering of BNIP3 and NIX and promote mitophagy progression (Marinković et al, 2021), although recent findings suggest that the role of their transmembrane domains in dimerization may be less critical than previously hypothesized (Bunker et al, 2023; Yamashita et al, 2025). Extending this concept, a recent study demonstrates that transmembrane SARs harbor intrinsically disordered regions (IDRs) with a net negative charge that are sufficient to drive organelle fragmentation, whereas the LC3-interacting regions (LIRs) are required for lysosomal delivery but are dispensable for organelle fragmentation. Remarkably, transplantation of these IDRs onto different organelle membranes can trigger fragmentation and autophagic turnover even at organelles distinct from their native location, revealing a modular and transferable mechanism for controlling organelle integrity and remodeling (Rudinskiy et al, 2025).

Not all transmembrane SARs are regulated by their constitutive turnover. ER-phagy SARs such as FAM134B, FAM134C, and CCPG1, as well as some mitophagy and Golgi SARs, are constitutively expressed and stably localized to the cytosolic face of their respective organelle membranes, even under non-stressed conditions. In these cases, the mechanisms that keep SARs inactive remain incompletely understood.

Excitingly, though, some activation mechanisms are beginning to be described. For example, FAM134B undergoes stress-induced ubiquitination on its reticulon homology domain (RHD), which promotes clustering of neighboring SARs (Foronda et al, 2023; González et al, 2023; Poveda-Cuevas et al, 2024). These ubiquitin moieties mediate interactions between adjacent RHDs that facilitate the formation of dense SAR assemblies that can induce membrane curvature and large-scale ER remodeling. Such clustering is likely to serve as a thresholding mechanism, ensuring that autophagosome formation is only triggered when SAR density—and thus organelle damage—is sufficiently high. At the same time, the SARs fragment the ER into bite-sized pieces small enough to be sequestered by autophagosomes (Bhaskara et al, 2019; Jiang et al, 2020).

## Recruiting the autophagy machinery: distinct modes of initiation

Despite their distinct spatial organization, both soluble and transmembrane SARs ultimately converge on the core autophagy machinery to initiate autophagosome formation. Soluble SARs, such as NDP52, TAX1BP1, OPTN and SQSTM1/p62, engage FIP200, a key scaffolding component of the ULK complex (Ravenhill et al, 2019; Turco et al, 2019; Vargas et al, 2019; Zhou et al, 2021). Direct binding of these SARs to the C-terminal Claw domain or a C-terminal region of the coiled-coil domain of FIP200 facilitates the recruitment of the ULK complex to the cargo, thereby initiating autophagosome nucleation (Ravenhill et al, 2019; Shi et al, 2020; Turco et al, 2019; Vargas et al, 2019). In addition to this, other proteins may also be recruited by soluble SARs: NDP52 and TAX1BP1 also recruit the TBK1 kinase via the NAP1 and SINTBAD adapters (Adriaenssens et al, 2025; Li et al, 2016; Ravenhill et al, 2019), and OPTN also directly recruits the kinase TBK1 (Li et al, 2016; Morton et al, 2008; Wild et al, 2011), which binds the PI3KC3-C1 (Nguyen et al, 2023). The PI3KC3-C1 may in turn recruit and activate the ULK complex (Chen et al, 2025). OPTN has additionally been shown to recruit ATG9A vesicles (Yamano et al, 2020). Notably, all these autophagy factors are traditionally considered to be upstream factors (ie, SAR recruiters) within hierarchy of autophagy cascade.

Transmembrane SARs, employ at least two distinct strategies to recruit the autophagy initiation machinery (Fig. 3). Some SARs, such as CCPG1 and FUNDC1, mimic soluble SARs by directly binding to FIP200, thereby facilitating recruitment of the ULK complex (Adriaenssens et al, 2025; Smith et al, 2018). Others act through a WIPI-dependent mechanism, in which the SAR recruits WIPI proteins that, in turn, interact with ATG13 to promote the assembly of the ULK complex (Adriaenssens et al, 2025; Bunker et al, 2023). This WIPI-mediated route appears particularly important in NIX/BNIP3-driven mitophagy and TEX264-dependent ER-phagy (Adriaenssens et al, 2025; An et al, 2019). The WIPI-dependent mechanism of mitophagy initiation has been most extensively investigated for BNIP3 and NIX. The two SARs bind to the top of the beta propeller between blades 2 and 3 of WIPI2 and WIPI3. This interaction, on the SAR side, uses two interaction domains: an extended stretch in the cytoplasmic unstructured regions of BNIP3 and NIX termed the minimal essential region (MER) (Adriaenssens et al, 2025; Bunker et al, 2023), and the canonical LIRs (Adriaenssens et al, 2025). The same binding site in WIPI2 between blades 2 and 3 occupied by the MER is also bound by ATG16L (Dooley et al, 2014; Strong et al, 2021). The involvement of the LIR motif in the interaction with WIPI2 is reminiscent of the involvement of this motif in the interaction of many SARs (such as p62, OPTN, BCL2L13, FUNDC1 and CCPG1) with the FIP200 Claw domain (Adriaenssens et al, 2025; North et al, 2025; Turco et al, 2019; Zhou et al, 2021). Mutation of the LIR motif in SARs may frequently therefore not only compromise ATG8 binding, but also hamper the recruitment of the upstream autophagy machinery.

BNIP3/NIX subsequently indirectly recruit the ULK complex, because after their recruitment by these receptors, WIPI2 and WIPI3 bind to two adjacent sites in the N-terminal part of the unstructured domain of ATG13. This interaction is required for BNIP3- and NIX-driven mitophagy but is dispensable for PINK1/Parkin-mediated mitophagy. Through this WIPI2/WIPI3–ATG13 interaction BNIP3 and NIX can indirectly recruit the ULK complex to the mitochondrial surface. FIP200-dependent mechanisms for ULK complex recruitment also exist in DFP-induced mitophagy. This DFP-initiated recruitment may be driven by the BCL2L13,

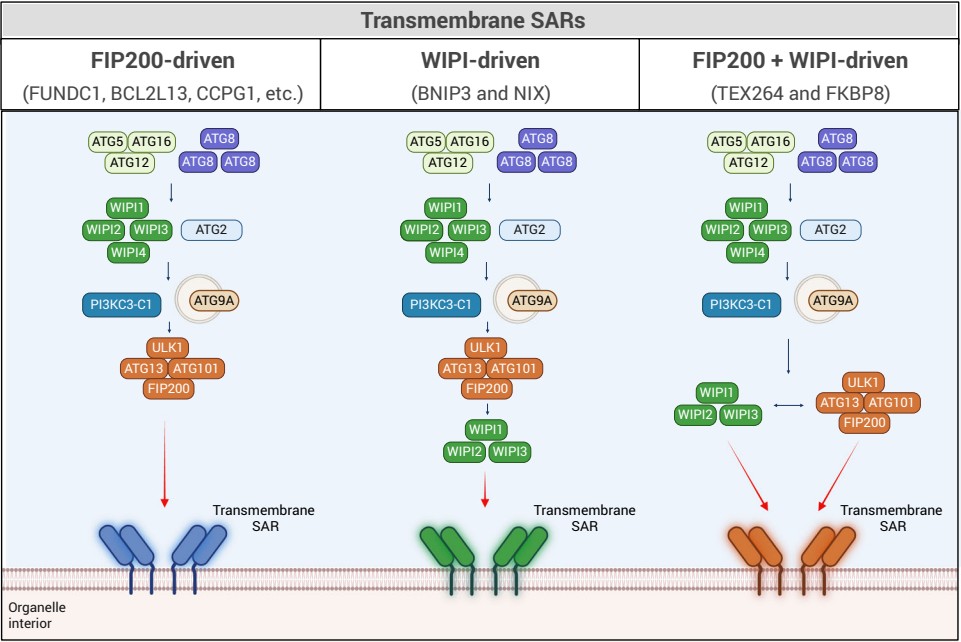

**Figure 3. Recruitment of the core autophagy machinery by transmembrane SARs.**

Transmembrane SARs engage the autophagy initiation machinery through at least three distinct mechanisms. Left, FIP200-driven SARs (e.g., FUNDC1, BCL2L13, CCPG1) interact directly with the FIP200 subunit of the ULK complex, facilitating its recruitment to cargo. Middle, WIPI-driven SARs (e.g., BNIP3 and NIX) recruit WIPI1, WIPI2, or WIPI3, which in turn engages ATG13 to promote ULK complex assembly at the cargo site without direct FIP200 engagement. Right, some SARs (e.g., TEX264 and FKBP8) combine features of both mechanisms, interacting with both FIP200 and WIPI proteins to ensure robust ULK complex recruitment. All pathways converge on the recruitment of downstream autophagy machinery, including ATG5-ATG12-ATG16, ATG8/LC3 lipidation, and ATG9A vesicle trafficking, ultimately leading to phagophore formation around the cargo. Reproduced, in part, from Adriaenssens et al, *Nature Cell Biology*, 27, 1272–1287 (2025), with permission.

FUNDC1 or FKBP8 proteins, which are also expressed in the cells investigated (Yamashita et al, 2025). Despite these differences, the process of autophagosome biogenesis downstream of indirect ULK complex recruitment may then proceed as other forms of autophagy do, as FIP200 and the activity of the PI3KC3-C1 are still required for cargo degradation. Downstream of the activities of the autophagy machinery, the LIR motifs of BNIP3 and NIX can interact with lipidated ATG8 proteins to keep the phagophore close to cargo (Yamashita et al, 2025). Further feedback loops may be involved, where ATG8 proteins recruit other LIR-containing autophagy machinery components to render the pathway more robust (North et al, 2025; Padman et al, 2019; Yamashita et al, 2025).

So why do we observe multiple pathways through which the core autophagy machinery is recruited to cargo? One possibility is that parallel pathways may confer robustness with respect to the diversity of stress inputs triggering autophagy and the process of autophagosome biogenesis. Alternatively, differences in the expression levels of autophagy proteins across tissues may favor the use of distinct mechanisms. Importantly, these recruitment mechanisms are not mutually exclusive and could occur simultaneously, potentially enhancing or optimizing autophagosome formation. From a conceptual point of view, it is interesting that a strict hierarchy for the productive recruitment of the autophagy machinery does not seem to exist. Instead, it appears that as long as a robust enough network of interactions among the autophagy machinery can be established, various ways to cast the net are possible.

# Kinase regulation of transmembrane SAR-initiated autophagy

Autophagy receptor activation and downstream signaling are frequently modulated by phosphorylation: a regulatory principle that was first uncovered in yeast through studies of SARs involved in mitophagy and pexophagy. Subsequently, phosphorylation-dependent regulation was also demonstrated for Atg19 in the Cvt pathway (Pfaffenwimmer et al, 2014; Tanaka et al, 2014). Building on these studies, kinase regulation of mammalian SARs was later revealed. For soluble SARs, TBK1 has emerged as a key regulator (Adriaenssens et al, 2022; Goodall et al, 2022; Vargas et al, 2023). TBK1 phosphorylates SARs such as p62, TAX1BP1, OPTN and NDP52, enhancing their ability to bind ubiquitin, FIP200 and ATG8 proteins, and therefore to initiate autophagosome formation (Heo et al, 2015; Moore and Holzbaur, 2016; Nguyen et al, 2023; Richter et al, 2016; Rogov et al, 2013; Thurston et al, 2009; Wild et al, 2011; Zhou et al, 2021). In some contexts, TBK1 also facilitates the indirect recruitment of ULK1 to damaged cargo via binding to the PI3KC3-C1, suggesting a flexible and multi-tiered role of the latter complex in autophagy initiation (Nguyen et al, 2023).

Phosphorylation is also an important regulatory modification for transmembrane SARs, regulating the interaction with autophagy factors such as BNIP3 and NIX with ATG8 proteins (Adriaenssens et al, 2025; Rogov et al, 2017; Zhu et al, 2013). In comparison to the soluble SARs, ULK1 appears to play a more prominent and non-redundant role in autophagy pathways

initiated by transmembrane SARs. For example, mitophagy mediated by BNIP3 and NIX is strictly dependent on ULK1 kinase activity as knockout of ULK1 or its inhibition with small molecules completely abolishes DFP-induced mitophagy (Adriaenssens et al, 2025; Wilhelm et al, 2022). It was also shown that AMPK inhibits BNIP3/NIX mitophagy by phosphorylating ULK1 at S554 and S694 resulting in its sequestration by 14-3-3 proteins (Longo et al, 2024). In contrast, TBK1 inhibitors do not affect BNIP3/NIX driven mitophagy (Adriaenssens et al, 2025; Longo et al, 2024). BCL2L13 seems to be regulated in a different manner by upstream kinases because it was shown that AMPK controls its TBK1-dependent phosphorylation at S275. This residue lies close to one of its LIR motifs and strengthens its binding to ATG8 proteins (Ahmed et al, 2025).

Beyond TBK1 and ULK1, several other kinases have been implicated in organelle-selective autophagy. The constitutively active kinase CK2 has been shown to regulate the ER-phagy SARs TEX264, FAM134B, and FAM134C (Berkane et al, 2023; Chino et al, 2022). Phosphorylation of TEX264 by CK2 enhances its affinity for ATG8 proteins (Chino et al, 2022), resembling the regulation of soluble SARs, which are frequently phosphorylated by TBK1 near their LIR motifs. However, contradictory findings have emerged for FAM134B: some studies identify CK2 as an essential activator of FAM134B- and FAM134C-dependent ER-phagy by promoting ubiquitination and clustering (Berkane et al, 2023), while others suggest it inhibits ATG8 binding (Di Lorenzo et al, 2022). Interestingly, CK2 has also been implicated in the regulation of mitophagy through FUNDC1, where its phosphorylation near the LIR motif silences FUNDC1. However, upon hypoxia, the phosphatase PGAM5 dephosphorylates FUNDC1 at S13, thereby strengthening LC3 binding and promoting mitophagy (Chen et al, 2014). These contrasting roles of CK2 as both an activator and inhibitor of transmembrane SARs could highlight cell type-specific differences, context-dependent feedback loops, or distinct roles of CK2 isoforms. More broadly, it raises an intriguing question: how does a constitutively active kinase like CK2 achieve the temporal and spatial specificity required to regulate a process such as selective autophagy?

Src-family kinases have been proposed to regulate mitophagy by phosphorylating the mitochondrial SAR FUNDC1, particularly under hypoxic conditions (Liu et al, 2012). Early studies suggested that Src-mediated phosphorylation inhibits FUNDC1, while phosphatase-mediated dephosphorylation promotes its interaction with ATG8s and mitophagy activation (Liu et al, 2012). However, recent work has shifted this perspective, establishing NIX and BNIP3 and not FUNDC1 as the principal facilitators of hypoxia-induced mitophagy (Clague and Urbé, 2025; Ganley and Simonsen, 2022; Küng et al, 2025). In line with this, a recent study demonstrated that Src-family kinase activity inhibits FUNDC1-mediated mitophagy during ischemia/reperfusion (I/R) injury by maintaining it in a phosphorylated, inactive state (Tang et al, 2024). This reinforces the view that FUNDC1 may be negatively regulated during certain types of mitochondrial stress. These findings raise the question of whether Src-family kinases function primarily as negative regulators of FUNDC1 or if their influence on mitophagy reflects broader shifts in cellular signaling networks during stress. Future work will be needed to clarify the context in which Src activity modulates mitophagy, and how this intersects with the emerging dominance of NIX/BNIP3 in hypoxia-related pathways.

# The unresolved question of substrate identity

Despite the recent progress in our understanding of how transmembrane SARs recruit the core machinery and become activated, we still know remarkably little about what specific cargo or damage signals these SARs respond to. This stands in contrast to soluble SARs, where cargo is typically marked by ubiquitin and recognition logic is relatively well established (Goodall et al, 2022). For many transmembrane SARs substrate specificity remains poorly defined. Why do cells assign different SARs to a single organelle, often with overlapping localization but seemingly distinct regulation? One possibility is that each SAR monitors a different type of organelle stress, such as protein misfolding, membrane defects, or disrupted contact sites. However, systematic evidence supporting such models is only beginning to emerge. Adding to this complexity, some transmembrane SARs exist in multiple isoforms, such as FAM134B, which can play selective roles during specific stages of differentiation or in distinct tissues (Buonomo et al, 2025). This highlights that the functional output of a given SAR may depend not only on the type of cargo or organelle stress, but also on isoform-specific properties and cellular context.

Recent work on ER-phagy has started to address this question. A comprehensive proteomic study in human neurons revealed that deletion of individual ER-phagy SARs, including FAM134A, FAM134B, FAM134C, TEX264, and CCPG1, led to the selective accumulation of distinct ER proteins, such as curvature-shaping REEP family members and contact site proteins like VAPA (Hoyer et al, 2024). While some luminal proteins appeared to be redundantly regulated, others were uniquely stabilized upon loss of specific SARs, pointing to discrete and non-overlapping substrate pools. In addition, TEX264 has been shown to mediate the degradation of topoisomerase 1 cleavage complexes upon replication fork stalling via nucleophagy, a specialized form of ER-phagy (Lascaux et al, 2024). These findings suggest that ER-phagy SARs may indeed act as sensors for different functional or structural perturbations within the ER and even the nucleus. How such sensing occurs remains largely obscure, but several possibilities can be envisioned. For example, changes in membrane composition or curvature could be detected directly through SAR transmembrane segments. Alternatively, luminal tails or dedicated binding partners could convey these cues, as suggested by calnexin-dependent ER-phagy, in which calnexin has been implicated in linking luminal protein folding stress to selective ER turnover (Forrester et al, 2019).

For mitophagy, we have a relatively good understanding of the upstream logic of PINK1/Parkin-driven mitophagy, which responds to mitochondrial depolarization or protein aggregation in the mitochondrial matrix (Narendra et al, 2008; Uoselis et al, 2023a). NIX/BNIP3-driven mitophagy can be triggered by DFP-induced iron chelation (Allen et al, 2013). This treatment results in activation of HIF1alpha signaling and subsequent upregulation of BNIP3 and NIX at the transcriptional level (Fig. 4). DFP interferes with the assembly of iron sulfur clusters, suggesting that it also induces mitochondrial damage (Allen et al, 2013; Kontoghiorghes et al, 2020). However, mitochondrial damage does not seem to be a prerequisite for BNIP3/NIX mitophagy, because functional mitochondria, as defined by their membrane potential and respiratory capacity, can also be targeted by this pathway (Longo et al, 2024). Furthermore, NIX mediates developmentally regulated mitophagy during erythrocyte and

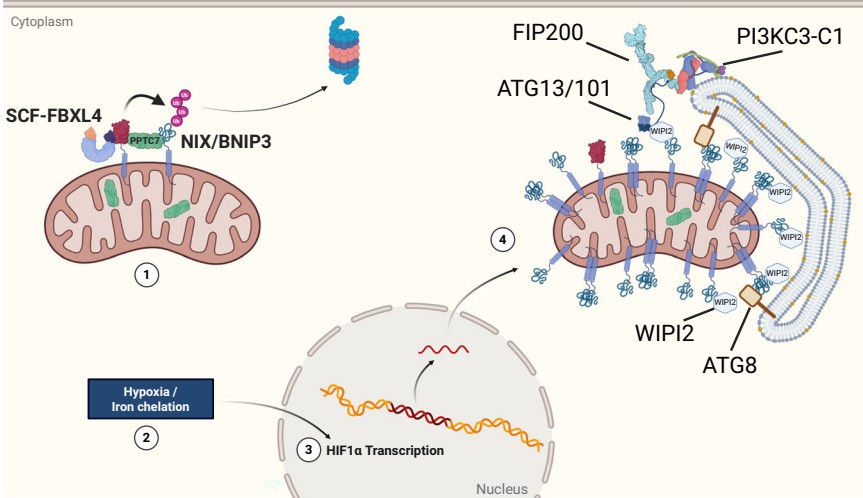

**Figure 4. Overview of upstream regulatory mechanisms and initiation of NIX/BNIP3-mediated mitophagy.**

(1) At steady state, NIX and BNIP3 levels at the mitochondrial outer membrane are regulated by SCF-FBXL4-mediated ubiquitination and subsequent proteasomal degradation, where the mitochondrial phosphatase PPTC7 acts as bridging factor between NIX/BNIP3 and the SCF-FBXL4 complex. (2) Upon iron chelation (e.g., by deferiprone (DFP) or hypoxia, stabilization of HIF1α triggers a transcriptional response. (3) HIF1α drives the expression of NIX and BNIP3, resulting in their accumulation at the mitochondrial surface. (4) NIX and BNIP3 initiate mitophagy by recruiting the autophagy machinery via WIPI proteins, which serve as adapters for ATG13 and promote assembly of the ULK complex. The subsequent recruitment of FIP200 and the PI3KC3-C1 complex enables phagophore formation and engulfment of mitochondria.

cardiomyocyte differentiation (Lampert et al, 2019; Mortensen et al, 2010; Sandoval et al, 2008). Unless these forms of mitophagy are triggered by developmentally regulated mitochondrial damage, these findings suggest that the functional state of the mitochondria is not a defining factor for BNIP3/NIX. Similarly, FUNDC1 has been shown to participate in mitophagy during cardiomyocyte differentiation (Lampert et al, 2019).

Much remains to be discovered about the cues that trigger BNIP3- and NIX-mediated mitophagy; even less is known about those that act through other mitophagy SARs, such as FUNDC1 (Liu et al, 2012), FKBP8 (Bhujabal et al, 2017) and BCL2L13 (Murakawa et al, 2015). Whether these cues include specific metabolic imbalances, membrane properties, or signaling events remains an open question. As long as the molecular stimuli that activate each SAR remain undefined, it will be difficult to define the logic with which cells survey the integrity of their organelles and engage selective autophagy in response to damage.

## Concluding remarks and perspectives

The pronounced differences between soluble and transmembrane SARs highlights the architectural and regulatory flexibility of the selective autophagy system. Both classes of SARs enable the targeted removal of damaged or surplus cellular components, yet they do so through distinct modes of cargo recognition, regulatory logic, and engagement with the autophagy initiation machinery. Soluble SARs rely heavily on ubiquitin recognition, TBK1-mediated phosphorylation, and dynamic recruitment, whereas transmembrane SARs are shaped by their fixed spatial context and rely more on ULK1 activity and context-dependent activation.

Within these broad frameworks, though, several key questions remain unanswered. What are the precise molecular mechanisms that prevent constitutively expressed transmembrane SARs from triggering basal autophagy? Are additional kinases or scaffolding proteins yet to be discovered that fine-tune autophagy initiation in an organelle-specific manner? And to what extent can this knowledge be harnessed to design targeted therapeutic strategies for diseases marked by organelle dysfunction, such as neurodegeneration, cancer, and metabolic syndromes?

Furthermore, engagement of certain transmembrane SARs does not always trigger macroautophagy: it can instead drive the selective removal of organelle portions via microautophagy or the formation of organelle-derived vesicles that fuse with lysosomes, as reported for mitochondria and the ER (Chino and Mizushima, 2023; König and McBride, 2024; Loi et al, 2019; Sakai et al; Towers et al, 2021; Uoselis et al, 2023b). What determines whether a damaged organelle fragment is targeted for macroautophagy, microautophagy, or vesicle-mediated degradation remains completely unknown, but is likely to represent another crucial layer of regulation.

As mechanistic insights continue to emerge—fueled by reconstitution experiments, structural studies, and in vivo models—it will be important to contextualize membrane anchoring not simply as a physical constraint, but as an active component of the regulatory logic that governs selective autophagy.

## Peer review information

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

## Acknowledgements

We thank members of the Martens lab for discussions and Elisabeth Holzer for help with the figures. The schematics were in part generated with BioRender.

## Author contributions

**Elias Adriaenssens**: Conceptualization; Funding acquisition; Visualization; Writing—original draft; Writing—review and editing. **Sascha Martens**: Conceptualization; Funding acquisition; Visualization; Writing—original draft; Writing—review and editing.

## Disclosure and competing interests statement

SM is a member of the scientific advisory board of Casma Therapeutics.

