## [Peer Review File · The EMBO Journal]

Mechanism of autophagy initiation by transmembrane selective autophagy receptors

Elias Adriaenssens and Sascha Martens

Corresponding author(s): Sascha Martens (sascha.martens@univie.ac.at) , Elias Adriaenssens (elias.adriaenssens@univie.ac.at)

Review Timeline:

Submission Date:	23rd Jul 25
Editorial Decision:	15th Aug 25
Revision Received:	11th Sep 25
Editorial Decision:	25th Sep 25
Revision Received:	28th Sep 25
Accepted:	6th Oct 25

Editor: William Teale

Transaction Report:

Dear Sascha,

Thank you for submitting your manuscript for consideration by the EMBO Journal. It has now been seen by two referees whose comments are enclosed. As you will see, both referees express interest in your manuscript and are in favour of publication, pending satisfactory minor revision.

Given the referees' positive recommendations, I would like to invite you to submit a revised version of the manuscript, addressing the comments of all three reviewers. I should add that it is EMBO Journal policy to allow only a single round of revision, and acceptance of your manuscript will therefore depend on the completeness of your responses in this revised version.

We generally allow three months as standard revision time. As a matter of policy, competing manuscripts published during this period will not negatively impact on our assessment of the conceptual advance presented by your study. However, we request that you contact the editor as soon as possible upon publication of any related work, to discuss how to proceed.

Thank you for the opportunity to consider your work for publication. I look forward to your revision.

Best wishes,

William

William Teale, PhD
Editor
The EMBO Journal
w.teale@embojournal.org

We realize that it is difficult to revise to a specific deadline. In the interest of protecting the conceptual advance provided by the work, we recommend a revision within 3 months (13th Nov 2025). Please discuss the revision progress ahead of this time with the editor if you require more time to complete the revisions. Use the link below to submit your revision:

Referee #1:

This is a very well written, clear and complete manuscript, which is topical and will optimally serve the field of autophagy

I have just minor comments aimed at improving the clarity and fluidity of the text, and the content.

Title. Although making it longer, it would be better to have a more specific title like for example "Mechanism of autophagy initiation by transmembrane selective autophagy receptors. The term transmembrane receptors is a bit too vague.

L47 and L81. To be precise, it may be misleading to write that entire organelles are targeted by selective autophagy because, as discussed in the review, organelles such as mitochondria, ER and peroxisomes undergoes fission/fragmentation and thus only part of them are degraded.

L48. The autophagy receptors are often called selective autophagy receptors and abbreviated SARs. I would suggest introducing this term and abbreviation here, and then use SAR/SARs throughout the entire manuscript instead of soluble receptors, transmembrane receptors, autophagy cargo receptors, autophagy receptors....so you will have soluble SARs, transmembrane SARs...simplifying the text/figures and adding consistency and clarity.

L83. I would change "...by selective autophagy, a pathway that..." into "...by selective type of autophagy, a set of pathways that..."

L90. I would change "...then fuses with a lysosome,..." into "...then fuses first with late endosomes to form an amphisome and subsequently with a lysosomes or directly with lysosomes,..."

L92. I would change "...core autophagy machinery..." into "...core autophagy-related (ATG) machinery...", so that the abbreviation ATG is introduced.

L92. I would change "...in all pathways..." into "...in most of the pathways..." since there are unconventional types of autophagy in which autophagy and autophagosome formation do not depend on the entire core ATG machinery. E.g., Atg5/Atg7-independent autophagy.

L94 and L140. I would change ULK1 complex into ULK complex because it seems that it can also contain ULK2.

L98. NRF2 must be added as a subunit of the PI3KC3-C1 complex.

L102. You may consider introducing the abbreviation WIPI.

L108. You could extend the sentence by saying the ATG12-ATG5-ATG16L1 is formed through the action of a second ubiquitin-like conjugation system.

L113. I would change ATG2A into ATG2 proteins. ATG2 and ATG2B have redundant functions in vivo and some of the studies in vitro show that ATG2B has the same lipid transfer activity as ATG2A. I would also modify Fig. 1B accordingly.

L120-121. I would change "...the ESCRT machinery converts..." into "...the ESCRT machinery appears to convert..." because this requirement is still a bit controversial and not totally clarified because there are several reports indicating that in the tested conditions, the ESCRT components are not required for autophagy.

L142. The abbreviations/families LC3s and GABARAPs have not been defined.

L153. You could also consider to mention that (most of) the soluble SARs are ubiquitin dependent while the transmembrane SARs are not.

L186. The glycogen receptors could be another example

L217-260. I think that when discussing about organelle fragmentation, it must be mentioned that in the case of mitochondria

(yeast and mammals) and peroxisomes (yeast), it has been shown that the machinery involved in the fission of this organelle interact with the ATG machinery and it is required for mitophagy/pexophagy.

I also think that in this context is important to describe here or elsewhere this very recent study:

<https://pubmed.ncbi.nlm.nih.gov/40760246/>

L333-340. Here or elsewhere, I would make clear for non-experts that the different ATG recruitment mechanism are not mutually exclusive and could occur simultaneously, possible enhancing/optimizing autophagosome formation induction.

L342. To be historically correct, it must be indicated that the regulation of SARs was initially discovered in yeast by studying the SARs involved in mitophagy and pexophagy. Later, also Atg19 in the Cvt pathway was shown to be regulated through phosphorylation.

L415-427. You could be a little more specific (although still speculative) by indicating that changes in the membrane may be sensed by the SARs transmembrane segments (it has been shown in other quality control pathway, e.g. for IRE1), their luminal tails or eventual binding partners. In this latter case, I think that there is the example of calnexin in ER-phagy.

L467-486. Another interesting aspect is that you could consider mentioning is that in some instances, engagement of the transmembrane SARs leads to the degradation of part of the organelles by microautophagy or organelle-derived vesicles that fuse with lysosomes. This has been shown for mitochondria and ER. What determines the targeting to macroautophagy, microautophagy or vesicles remain totally unknown.

Referee #2:

The manuscript is an excellent and well-crafted review, offering a thorough and insightful dissection of the role of transmembrane receptors in autophagy initiation. It elegantly integrates recent discoveries with a critical analysis, raising scientifically significant questions and highlighting key issues likely to shape the direction of future research in the field. The text is well-organized, scientifically rigorous, and engaging to read. I have only a few minor comments that may help improve clarity and presentation:

Figure 1 - The label for the ATG8 ligation complex is currently difficult to read. I recommend increasing font size, applying a consistent color code for improved visual distinction, and clarifying the label for PI(3)P, which is not entirely legible.

Line 120 - Please provide a brief description of ESCRT (Endosomal Sorting Complex Required for Transport), as this acronym may not be familiar to all readers.

Line 186 - Similarly, please define zinosome for the benefit of readers who may not be familiar with this term.

Line 233 - I suggest adding a reference before the full stop to substantiate the statement.

Figure 4 - Consider reducing the relative size of the nucleus and enlarging the two mitochondria so that the labels for LC3, WIPI2, and PPTC7 are more easily readable; at present, these elements appear too small.

Additional point for discussion - It may be worth mentioning that some of the receptors discussed in the manuscript exist in different isoforms, such as FAM134B. These isoforms can display selective roles during specific stages of cellular differentiation, for example during myogenesis (doi.org/10.1038/s44318-024-00356-2). Including this aspect would underscore the complexity of receptor-mediated regulation and could provide a framework for understanding how their function may vary in a time-dependent or subcellular localization-dependent manner.

Point-by-point reply**Referee #1:**

This is a very well written, clear and complete manuscript, which is topical and will optimally serve the field of autophagy. I have just minor comments aimed at improving the clarity and fluidity of the text, and the content.

We thank the reviewer for the kind words and greatly appreciate the time and effort the reviewer dedicated to carefully review our manuscript and to provide the excellent suggestions below.

Title. Although making it longer, it would be better to have a more specific title like for example "Mechanism of autophagy initiation by transmembrane selective autophagy receptors. The term transmembrane receptors is a bit too vague.

We thank the reviewer for this excellent suggestion and have incorporated it.

L47 and L81. To be precise, it may be misleading to write that entire organelles are targeted by selective autophagy because, as discussed in the review, organelles such as mitochondria, ER and peroxisomes undergoes fission/fragmentation and thus only part of them are degraded.

We agree with the reviewer that 'entire' may overstate what happens in cells. We have therefore modified L47 and L81.

L48. The autophagy receptors are often called selective autophagy receptors and abbreviated SARs. I would suggest introducing this term and abbreviation here, and then use SAR/SARs throughout the entire manuscript instead of soluble receptors, transmembrane receptors, autophagy cargo receptors, autophagy receptors....so you will have soluble SARs, transmembrane SARs...simplifying the text/figures and adding consistency and clarity.

We apologize for the confusion our phrasing in the submitted manuscript may have caused but agree that soluble and transmembrane SARs would simplify the text and improve its readability considerably. We have therefore adopted this abbreviation.

L83. I would change "...by selective autophagy, a pathway that..." into "...by selective type of autophagy, a set of pathways that..."

We thank the reviewer for this suggestion and have incorporated it.

L90. I would change "...then fuses with a lysosome,..." into "...then fuses first with late endosomes to form an amphisome and subsequently with a lysosomes or directly with lysosomes,..."

We thank the reviewer for this suggestion and have incorporated it.

L92. I would change "...core autophagy machinery..." into "...core autophagy-related (ATG) machinery...", so that the abbreviation ATG is introduced.

We thank the reviewer for this suggestion and have incorporated it.

L92. I would change "...in all pathways..." into "...in most of the pathways..." since there are unconventional types of autophagy in which autophagy and autophagosome formation do not depend on the entire core ATG machinery. E.g., Atg5/Atg7-independent autophagy.

We thank the reviewer for this suggestion and have incorporated it.

L94 and L140. I would change ULK1 complex into ULK complex because it seems that it can also contain ULK2.

We thank the reviewer for this suggestion and have incorporated it.

L98. NRF2 must be added as a subunit of the PI3KC3-C1 complex.

We thank the reviewer for this suggestion and have incorporated 'NRBF2 as the fifth subunit' in our manuscript.

L102. You may consider introducing the abbreviation WIPI.

We thank the reviewer for this suggestion and have incorporated it.

L108. You could extend the sentence by saying the ATG12-ATG5-ATG16L1 is formed through the action of a second ubiquitin-like conjugation system.

We thank the reviewer for this suggestion and have incorporated it.

L113. I would change ATG2A into ATG2 proteins. ATG2 and ATG2B have redundant functions in vivo and some of the studies in vitro show that ATG2B has the same lipid transfer activity as ATG2A. I would also modify Fig. 1B accordingly.

We thank the reviewer for this suggestion and have modified our text accordingly.

L120-121. I would change "...the ESCRT machinery converts..." into "...the ESCRT machinery appears to convert..." because this requirement is still a bit controversial and not totally clarified because there are several reports indicating that in the tested conditions, the ESCRT components are not required for autophagy.

We thank the reviewer for this suggestion and have modified our text accordingly.

L142. The abbreviations/families LC3s and GABARAPs have not been defined.

We thank the reviewer for this suggestion and have incorporated it.

L153. You could also consider to mention that (most of) the soluble SARs are ubiquitin dependent while the transmembrane SARs are not.

We thank the reviewer for this suggestion and have modified our text accordingly.

L186. The glycogen receptors could be another example

We thank the reviewer for this suggestion and have incorporated it.

L217-260. I think that when discussing about organelle fragmentation, it must be mentioned that in the case of mitochondria (yeast and mammals) and peroxisomes (yeast), it has been shown that the machinery involved in the fission of this organelle interact with the ATG machinery and it is required for mitophagy/pexophagy.

I also think that in this context is important to describe here or elsewhere this very recent study: <https://pubmed.ncbi.nlm.nih.gov/40760246/>

We thank the reviewer for this suggestion and have incorporated it. We also added the recently published work from the Molinari lab.

L333-340. Here or elsewhere, I would make clear for non-experts that the different ATG recruitment mechanism are not mutually exclusive and could occur simultaneously, possible enhancing/optimizing autophagosome formation induction.

We thank the reviewer for this suggestion and have incorporated it.

L342. To be historically correct, it must be indicated that the regulation of SARs was initially discovered in yeast by studying the SARs involved in mitophagy and pexophagy. Later, also Atg19 in the Cvt pathway was shown to be regulated through phosphorylation.

We thank the reviewer for this suggestion and have incorporated it.

L415-427. You could be a little more specific (although still speculative) by indicating that changes in the membrane may be sensed by the SARs transmembrane segments (it has been shown in other quality control pathway, e.g. for IRE1), their luminal tails or eventual binding partners. In this latter case, I think that there is the example of calnexin in ER-phagy.

We thank the reviewer for this suggestion and have incorporated it.

L467-486. Another interesting aspect is that you could consider mentioning is that in some instances, engagement of the transmembrane SARs leads to the degradation of part of the organelles by microautophagy or organelle-derived vesicles that fuse with lysosomes. This has been shown for mitochondria and ER. What determines the targeting to macroautophagy, microautophagy or vesicles remain totally unknown.

We thank the reviewer for this suggestion and have incorporated it.

Referee #2:

The manuscript is an excellent and well-crafted review, offering a thorough and insightful dissection of the role of transmembrane receptors in autophagy initiation. It elegantly integrates recent discoveries with a critical analysis, raising scientifically significant questions and highlighting key issues likely to shape the direction of future research in the field. The text is well-organized, scientifically rigorous, and engaging to read. I have only a few minor comments that may help improve clarity and presentation:

We thank the reviewer for the kind words and greatly appreciate the time and effort the reviewer dedicated to carefully review our manuscript and to provide the excellent suggestions below.

Figure 1 - The label for the ATG8 ligation complex is currently difficult to read. I recommend increasing font size, applying a consistent color code for improved visual distinction, and clarifying the label for PI(3)P, which is not entirely legible.

We thank the reviewer for this suggestion. We have modified the figure accordingly.

Line 120 - Please provide a brief description of ESCRT (Endosomal Sorting Complex Required for Transport), as this acronym may not be familiar to all readers.

We thank the reviewer for this suggestion and have incorporated it.

Line 186 - Similarly, please define zincosome for the benefit of readers who may not be familiar with this term.

We thank the reviewer for this suggestion and have incorporated it.

Line 233 - I suggest adding a reference before the full stop to substantiate the statement.

We thank the reviewer for this suggestion and have incorporated it.

Figure 4 - Consider reducing the relative size of the nucleus and enlarging the two mitochondria so that the labels for LC3, WIPI2, and PPTC7 are more easily readable; at present, these elements appear too small.

We have changed the figure accordingly.

Additional point for discussion - It may be worth mentioning that some of the receptors discussed in the manuscript exist in different isoforms, such as FAM134B. These isoforms can display selective roles during specific stages of cellular differentiation, for example during myogenesis (doi.org/10.1038/s44318-024-00356-2). Including this aspect would underscore the complexity of receptor-mediated regulation and could provide a framework for understanding how their function may vary in a time-dependent or subcellular localization-dependent manner.

We thank the reviewer for this suggestion and have incorporated it.

Dear Sascha,

I have now had a chance to make some editorial suggestions on your review and have marked up the attached word document. Please feel free to take or leave them as you see fit, just return the final version through the link below.

I notice that you do not acknowledge any funding, please feel free to do so if you want to.

Best wishes,

William

William Teale, PhD
Editor
The EMBO Journal
w.teale@embojournal.org

We realize that it is difficult to revise to a specific deadline. In the interest of protecting the conceptual advance provided by the work, we recommend a revision within 3 months (24th Dec 2025). Please discuss the revision progress ahead of this time with the editor if you require more time to complete the revisions. Use the link below to submit your revision:

All editorial and formatting issues were resolved by the authors.

Dear Sascha,

I am pleased to inform you that your manuscript has been accepted for publication in the EMBO Journal.

Thank you for contributing a very interesting and useful review!

Your manuscript will be processed for publication by EMBO Press. It will be copy edited and you will receive page proofs prior to publication. Please note that you will be contacted by Springer Nature Author Services to complete licensing information.

Yours sincerely,

William

William Teale, PhD
Editor
The EMBO Journal
w.teale@embojournal.org
